# There Is Strength in Numbers: Quantitation of Fc Gamma Receptors on Murine Tissue-Resident Macrophages

**DOI:** 10.3390/ijms222212172

**Published:** 2021-11-10

**Authors:** Christof Vorsatz, Niklas Friedrich, Falk Nimmerjahn, Markus Biburger

**Affiliations:** 1Division of Genetics, Department of Biology, Friedrich-Alexander University Erlangen-Nürnberg, 91058 Erlangen, Germany; christof.vorsatz@fau.de (C.V.); nik.f.friedrich@fau.de (N.F.); falk.nimmerjahn@fau.de (F.N.); 2Medical Immunology Campus Erlangen, Friedrich-Alexander University Erlangen-Nürnberg, 91052 Erlangen, Germany

**Keywords:** Fc receptors, antibodies, macrophages, alveolar macrophages, interstitial macrophages, Kupffer cells, splenic macrophages, kidney resident macrophages, microglia, Langerhans cells, dermal macrophages

## Abstract

Many of the effector functions of antibodies rely on the binding of antibodies/immune complexes to cellular Fcγ receptors (FcγRs). Since the majority of innate immune effector cells express both activating and inhibitory Fc receptors, the outcome of the binding of immune complexes to cells of a given population is influenced by the relative affinities of the respective IgG subclasses to these receptors, as well as by the numbers of activating and inhibitory FcγRs on the cell surface. A group of immune cells that has come into focus more recently is the various subsets of tissue-resident macrophages. The central functions of FcγRs on tissue macrophages include the clearance of opsonized pathogens, the removal of small immune complexes from the circulation and the depletion of antibody-opsonized cells in the therapy of autoimmunity and cancer. Despite these essential functions of FcγRs on tissue-resident macrophages, an in-depth quantification of FcγRs is lacking. Thus, the aim of our current study was to quantify the various Fcγ receptors on macrophages in murine liver, lung, kidney, brain, skin and spleen. Our study identified a pronounced heterogeneity between FcγR expression patterns of the different tissue macrophages, which may reflect their specialized functions within their unique niches in different organ environments.

## 1. Introduction

Macrophages (MΦ) form a network of heterogeneous immunologic sentinels in all tissues, where they perform niche and tissue specific functions. This involves classical immune functions like combating infections [1], the resolution of inflammation [2,3], surveillance against tumors [4,5] and mediation of antibody-dependent antitumoral [6] or cell-depleting [7] effector functions. In addition, besides their immunologic functions, tissue-resident macrophages play pivotal roles in tissue homeostasis [8,9,10] and can modulate angiogenesis and lymphangiogenesis [11,12]. However, macrophages are also critically involved in various disorders, like cardiovascular [13] and pulmonary [14] diseases, type 2 diabetes [15] and cancer, where they can promote cancer initiation, malignant progression, tumor cell migration, in- and intravasation and the suppression of antitumor immunity (see, e.g., Reference [16]). Most tissue macrophages—with the exception of intestinal MΦ, which are constantly seeded by bone marrow-derived monocytes [17,18]—are derived from yolk sacs or fetal liver precursors that persist into adulthood as resident populations that are maintained by in situ proliferation [19,20,21,22,23,24]. After birth, bone marrow-derived monocytes can replenish certain tissue-resident MΦ following tissue injury, infection and inflammation (reviewed, e.g., in Reference [25]). Among the innate immune effector cells, macrophages express the broadest set of cell surface receptors, including pattern recognition receptors and complement and Fc receptors, allowing them to directly or indirectly detect pathogens. The family of murine FcγR includes the inhibitory FcγRIIb (CD32b) and the activating receptors FcγRI, III and IV. Whereas FcγRIIb modulates signaling via an intracellular immunomodulatory tyrosine-based inhibitory motif (ITIM) in the same polypeptide chain as the ligand-binding domain, all activating FcγR signal via an activating ITAM motif provided by the accessory Fc receptor gamma chain [26]. While immune complexes (IC) can bind to all murine FcγR, including the low/medium affinity receptors FcγRIIb, FcγRIII and FcγRIV, monomeric IgG exclusively binds to high-affinity FcγRI (reviewed, e.g., in References [27,28]) and may be occupied by circulating IgG, at least in the blood. A central function of FcγRs on macrophages is to promote the clearance of opsonized pathogens in the periphery [29,30]. Likewise, opsonized host cells—for instance, cells that express viral antigens upon infection—can be killed via antibody-dependent cell-mediated cytotoxicity (ADCC) [31]. Along the same lines, macrophages have been identified as key effector cells for removing opsonized target cells in a variety of mouse model systems (e.g., References [6,7,32,33]). Apart from ADCC and phagocytosis, FcγRs on macrophages may also be involved in modulating MΦ polarization. Whereas the engagement of activating FcγRs induces the production of proinflammatory cytokines, the binding of immune complexes together with an engagement of Toll-like receptors (TLR) has been shown to drive the polarization of macrophages to a so-called M2b phenotype, also referred to as “regulatory macrophages”. These macrophages produced high levels of the immunosuppressive cytokine IL-10 and downmodulate TLR-induced IL-12 production [34,35]. Of note, with respect to infection, the antibody-mediated triggering of FcγR may have contrasting effects: On the one hand, the opsonization of pathogens that normally evade degradation in lysosomes like *Legionella pneumophila* and *Mycobacterium bovis* bacillus Calmette–Guérin results in targeting them at lysosomes, where they are efficiently eliminated [30]. On the other hand, the opsonization of pathogens like dengue virus with antibodies may cause increased uptake via FcγR engagement, resulting in antibody-enhanced infection [36,37,38].

In summary, IgG binding to FcγRs on macrophages is responsible for a wide range of effector functions underlying the activity of pathogen, autoreactive and therapeutic IgG antibody species. Importantly, the net effect of FcγR engagement by IgG immune complexes on any given macrophage is determined by the balance between signals from activating ITAM-bearing FcγRs and inhibitory ITIM-bearing FcγRIIb. This balance can be influenced by the size, IgG valency and subtype composition (due to different affinities between the different receptors and IgG subtypes) of the triggering immune complex but, obviously, also by the respective number of receptors on the cell surface [39,40,41,42].

Various groups have presented qualitative information on which Fcγ receptors are expressed on the various murine tissue macrophages and, often, data on mRNA expression via quantitative PCR and/or semi-quantitative flow cytometry data on the surface expression of murine FcγRs (see, e.g., References [43,44,45,46,47,48,49,50]). However, the relative mRNA expression levels of the different FcγRs do not necessarily translate one-to-one into ratios of receptor numbers on the cell surface. Data on the fluorescence intensities upon the binding of fluorescently labeled antibody conjugates are very valid for the comparison of relative expression levels of the same antigen on different cells or under different physiological conditions. However, as described in more detail in Reference [42], mainly due to inherent variations between the different antibody conjugates with respect to their specific fluorescence (i.e., fluorescence intensity per molecule), the quantitative comparison of different receptors is hampered. In addition, both methods do not provide any information on the actual receptor quantities on the cell surface.

Thus, as an in-depth characterization of the FcγR numbers on tissue-resident macrophages is lacking, we set out to quantify FcγRs on tissue-resident MΦ in the lung, liver, skin, spleen, kidney and brain. Our study identified a striking organ-specific expression pattern of FcγRs on different tissue-resident macrophage subsets, providing the basis for understanding their activity in triggering local IgG subclass-dependent effector functions.

## 2. Results

### 2.1. Identification of Organ Resident Macrophage Subsets

A prerequisite to quantifying FcγR expression on tissue-resident macrophage subsets is a reliable macrophage identification in tissue extracts of the skin, lung, liver, spleen, kidney and brain via a FACS analysis. As depicted in Figure 1, and in gating strategies provided for the different organs, most macrophage populations can be clearly identified based on the expression of selective marker subsets. However, certain organ-resident macrophage subsets, such as macrophage subsets in the lung and Kupffer cells in the liver, may be more difficult to separate, prompting us to include our gating strategies in each figure.

### 2.2. Impact of Organ Digestion on FcγR Expression

In order to isolate cells from tissue, mechanical disintegration may be insufficient to release cells. Thus, many groups have established protocols using enzymatic tissue digestion to increase the amount of tissue-resident immune cells. These protocols often include proteases for or the disruption of connective tissue components, which might affect the proteins on the cell membrane, including FcγRs. Thus, we first set out to verify that immune cell preparation by tissue digestion does not significantly affect the Fcγ receptor expression (Figure 2). We compared the FcγR expression on alveolar macrophages (AM) isolated either by bronchoalveolar lavage or by lung tissue digestion. As shown in Figure 2A, neither cell isolation protocol resulted in major differences in FcγR expression. In a further control experiment, we incubated leukocytes from the peripheral blood with the enzyme mix used for tissue disruption and again compared the FcγR expression to untreated cells. As shown in Figure 2B, no major effect of the FcγR expression on the different cell subsets was noted, suggesting that the enzyme mix used for tissue-resident macrophage isolation does not affect the cellular FcγR expression. Moreover, we established reference curves for each FcγR to allow a quantification of the FcγR numbers or antibody-binding sites for FcγR-specific antibodies (exemplarily shown for a quantification of FcγRI in Figure 2C).

### 2.3. FcγR Quantification on Lung Macrophage Subpopulations

In the lung, alveolar macrophages (AM) could be clearly identified as Siglec F^high^ CD11c^high^ cells as the most prominent myeloid population in the bronchoalveolar lavage (Figure 3A). Apart from AM, interstitial macrophages (IM) are the other major macrophage subset in the lung and are characterized by the expression of MerTK, CD11b and CD64 and the absence of Siglec F (reviewed in Reference [51]). To identify IM, we followed the strategy of Gibbings et al. [52] by gating on MerTK- and CD64-positive cells from enzymatically digested lung cell preparations, followed by the distinction between AM and IM via CD11c and CD11b (Figure 3B). However, for an accurate quantification of the FcγR numbers, a gating strategy identifying IM without CD64 is critical, as a fluorescence minus one (FMO) control is indispensable to correct for the relatively strong autofluorescence of certain macrophage subsets. To achieve this, we analyzed IM identified in a CD64-stained sample for other pairs of markers (CD11b vs. CD45, Siglec F vs. CD11c and F4/80 vs. MerTK) and their light scatter characteristics to allow a CD64-independent IM identification (Figure 3B). Based on the location of IM in these dot plots, we defined four corresponding gates. Applying the combination of these four gates on the single viable lineage^−/low^ cells clearly identified a MerTK^+^ CD64^+^ population and, thus, could be used for IM identification without staining for CD64.

Using this gating strategy, the quantification of FcγR numbers or, more accurately, the FcγR-specific antibody-binding capacity (ABC; represented as antibody-binding sites/cell) on the pulmonary MΦ subsets demonstrated a pronounced heterogeneity with respect to the FcγR numbers. Whereas AM showed a strong expression of all four FcγRs at 75,000–100,000 ABC, with FcγRIII being expressed at the highest level, the IM expressed FcγRI, IIb and III at much lower levels (roughly at 25,000 ABC). Of note, whereas the IM population showed a homogenous expression for FcγRI, IIb and III, it revealed two subpopulations with either a very high (120,000 ABC) or very low/absent FcγRIV expression (Table 1 and Figure 3C), consistent with the previous identification of several subsets of IM being present in the lung [52].

### 2.4. FcγR Quantification on Splenic Macrophage Subsets

The spleen represents another organ that contains numerous subsets of organ-resident macrophages, including red pulp macrophages (RPM) and several marginal zone macrophage subsets. RPM were identified by low-to-moderate CD11b and F4/80 expression, whereas the macrophages in the marginal zone lacked F4/80 but expressed CD169 (metallophilic macrophages) or SIGNR1 (marginal zone macrophages) (Figure 4A). With respect to FcγR numbers, RPM had a moderate expression of all the activating FcγRs and a low expression of the inhibitory FcγRIIb. In contrast, macrophages located in the marginal zone only revealed a moderate expression of FcγRIII (roughly 27,000 ABC) and low levels of FcγRIV (9000–12,000 ABC). The FcγRI and inhibitory FcγRIIb expression were even lower, with levels at around 3000 ABC (Table 1 and Figure 4B). The main difference between the RPM and marginal zone macrophages—the more pronounced expression of FcγRI and RIV on the former—was verified by a fluorescence microscopic analysis of spleen sections where both receptors were clearly detectable in the red pulp but barely outside (Figure 4C).

### 2.5. FcγR Quantification on Skin Resident Macrophage/DC Subsets

In the epidermis and dermis of the skin, we differentiated between dermal macrophages (DM) and Langerhans cells. The latter were identified as CD45^+^ CD11b^+^ F4/80^+^ Langerin^+^ in the epidermal cell preparations (Figure 5A). In the dermis, we identified two subsets of CD45^+^ F4/80^+^ Langerin^−^ CD11b^high^ cells, which differed in the expression of both in MHC II and CD11c (Figure 5B). With reference to the markers used both in this work and previous publications, the MHC II^−^ CD11c^−^ subpopulation correlated very well with both the DM characterized as MHC II^low^ CD11c^−^ CD11b^high^ F4/80^+^ by Merad et al. [53] and one of the two subsets of dermal macrophages described by Tamoutounour and colleagues (“P4”; MHC II^−^ CD11c^−^ CD11b^+^ FcγRI^high^). A microscopic analysis of the sorted MHCII^−^ CD11c^−^ DM indicated that this population also contained melanophages, i.e., macrophages that have taken up melanosomes from neighboring melanocytes. The cell surface markers of the MHC II^+^ CD11c^low to +^ subpopulation resembled—except for the extent of the CD11c expression—that of the second DM subset described by Tamoutounour et al. (“P5”; MHC II^+^, CD11c^− to low^ CD11b^+^ FcγRI^high^) [54], as well as that of the dermal Langerin-negative DCs described by Merad et al. (MHC II^+^ CD11c^+^ CD11b^high^ F4/80^+^) [53]. On the Langerhans cells, FcγR expression was restricted to roughly equal amounts (50,000 ABC) of the inhibitory FcγRIIb and the activatory FcγRIII. In contrast, on both subsets of CD45^+^ F4/80^+^ Langerin^−^ CD11b^high^ cells, both activating FcγRI (30,000 and 100,000 ABC) and FcγRIII (about 150,000 ABC) were expressed at high levels, while the expression of FcγRIV was lacking. Of further note, the inhibitory FcγRIIb showed a very high expression on the MHC II^+^ subpopulation of these cells (>200,000 ABC) and an extremely high expression with more than 600,000 ABC on the MHC II^−^ skin-resident macrophage subset (Table 1 and Figure 5C).

### 2.6. FcγR Quantification on Liver Resident Macrophages

In the liver, the major macrophage subset is Kupffer cells. Of note, however, when we stained liver the single-cell preparations for Kupffer cell markers F4/80 and Tim4 in combination with an FcγRIIb-specific antibody, we found two F4/80 and Tim4-double-positive subpopulations that differed markedly with respect to FcγRIIb expression (Figure 6A). Whereas CD31^+^ CD102^+^ SSC^high^ cells represented a population with very high FcγRIIb (CD32b) expression (Figure 6A, central panel), the nonendothelial population (CD31^−^ CD102^−^ SSC^low^) moderately expressed FcγRIIb (right panel). In addition, we noted that their relative amount varied to a great extent in the individual preparations. Of note, liver sinusoidal endothelial cells (LSEC) are known to express high levels of FcγRIIb [55]. Moreover, Lynch et al. showed that Kupffer cell preparations may contain a pronounced fraction of LSEC that mimic Kupffer cells with respect to cell surface marker expressions [56] due to a tight binding of the Kupffer cell membrane to the LSEC surface. To account for this issue, we added endothelial markers CD31 and/or CD102 to our FACS panel and found that, among the F4/80 Tim4-double-positive cells, the CD31^+^ CD102^+^ cells had a high side scatter (SSC^high^), which reflected a high granularity/complexity and very high FcγRIIb expression. In contrast, the nonendothelial CD31^−^ CD102^−^ SSC^low^ cells showed a moderate FcγRIIb expression (Figure 6A). This is in accordance with the data from Ganesan et al. [55] and Kumar et al. [46], showing that FcγRIIb expression on LSEC is much higher than on KC. To provide further evidence that the FcγRIIb-high subset is not a macrophage subset, we made use of BL6 Rosa26-*td tomato* x BL6 *cx3cr1-cre* mice, which express Cre recombinase under control of the cx3cr1 promoter and carry a floxed STOP cassette in front of the *td tomato* gene (Figure 6B). Due to the Cre-mediated excision of this STOP cassette, tdTomato is expressed in all cells that express CX3CR1—also transiently—at any point of their development, independent of an ongoing CX3CR1 expression. Thus, all liver resident Kupffer cells should express tdTomato in these mice. Indeed, we could verify that, among the F4/80^+^ Tim4^+^ cells, the SSC^high^ CD102^+^ subpopulation lacked tdTomato expression (Figure 6C). This strongly suggests that this FcγRIIb^high^ population is neither a myeloid cell population expressing endothelial markers nor are these cells aggregates of KC and endothelial cells. In fact, it seems much more likely that the FcγRIIb^high^ CD102^+^ cells are a population of liver endothelial cells (i.e., LSECS) that carry markers of macrophage membranes on their surfaces. In contrast, F4/80^+^ Tim4^+^ CD102^−^ SSC^low^ cells were clearly identified as tdTomato-expressing myeloid cells and represent the Kupffer cell population among F4/80^+^ Tim4^+^ cells. With respect to FcγR expression, the Kupffer cells expressed all three activating FcγRs, although FcγRI and FcγRIII were less abundant (around 20,000 ABC) than FcγRIV (50,000 ABC). Similar to the marginal zone and metallophilic macrophages in the spleen, the inhibitory FcγRIIb was expressed at relatively low levels (9000 ABC) (Table 1 and Figure 6E).

### 2.7. FcγR Quantification on Kidney Macrophages

In the kidney, we identified macrophages as CX3CR1-positive (by using CX3CR1-GFP reporter mice), F4/80^+^, FSC^high^ and Ly6C^−^ cells (Figure 7A). When analyzing the F4/80 expression and cell size of CD45 and GFP-double-positive cells, two populations were identified that were both F4/80-positive but differed prominently in cell size, as reflected by their forward light scatter characteristics (FSC: Figure 7A, right panel). The population with a larger cell size (FSC^high^) revealed a higher fluorescence with respect to GFP, as well as staining for CD11b, CD45 and F4/80 and for all Fcγ receptors upon the subsequent FcR expression analysis (not shown). Due to their very small size and the fact that they mirrored the antibody staining pattern of the FSC^high^ population but with lower fluorescence intensities, we considered the CD45^+^ CX3CR1-GFP^+^ CD11b^+^ F4/80^+^ Ly6C^−^ FSC^high^ cells to be kidney-resident macrophages and assumed that the FSC^low^ population may represent subcellular particles thereof, which were not analyzed further. The quantification of FcγRs revealed that all the FcγR species were expressed. Among the activating FcγRs, FcγRIV was expressed at the highest level (200,000 ABC), whereas FcγRI (59,000 ABC) and FcγRIII (38,000 ABC) were present at lower levels (Figure 7B and Table 1). With 95,000 ABC, the inhibitory FcγRIIb was expressed at relatively high levels compared to the other organ-resident macrophage subsets (with the exception of the dermal macrophages) (Table 1).

### 2.8. FcγR Quantification on Brain Macrophages

In the brain, microglia represent the predominant organ resident macrophage subset. By using CX3CR1-GFP reporter mice and CD45, we were able to clearly identify microglia in brain single-cell preparations, which revealed intermediate CD45 expression and very high *cx3cr1*-associated GFP fluorescence (Figure 8A). They also displayed a pronounced expression of CD11b (not shown). While the high-affinity FcγRI and the inhibitory FcγRIIb were expressed at roughly equal levels (50,000 ABC), FcγRIII was much less abundant (28,000 ABC), and FcγRIV was expressed at very low levels during the steady state (3000 ABC) (Figure 8B and Table 1).

## 3. Discussion

### 3.1. Importance of FcγR Quantification on Tissue-Resident Macrophage Subsets

Studies by many groups over the last decades have highlighted that organ-resident macrophages play a decisive role for tissue homeostasis and for the orchestration of tissue-specific immune responses. The notion that mostly tissue-resident macrophages and not NK cells are the major effector cell subset in mediating the cytotoxic or immunomodulatory antibody activity underscores that an in-depth understanding of FcγR expression on tissue-resident macrophages is of critical importance. In addition, to understand the involvement of tissue-resident macrophages as key effector cells for autoantibody-dependent inflammation or the destruction of organs, including the skin, liver, lung and kidney, also requires a thorough analysis of FcγR expression on tissue-resident macrophages. As different IgG subclasses show different affinities for individual activating and inhibitory FcγR, knowledge about the actual number of individual FcγRs present on organ-resident macrophages also allows the mathematical modeling of (auto)antibody activity and, hence, may allow to predict therapeutic, antiviral and autoantibody activity in the future [41]. To provide this lacking set of data, we quantified FcγR expression on organ-resident macrophages of the skin, lung, liver, kidney, brain and spleen of mice. Our study emphasizes that both the expression pattern, as well as the actual number of FcγRs present on the surface of different organ-resident macrophages, differ dramatically.

More importantly, our set of data is in line with many results obtained in different in vivo model systems of (auto)antibody activity.

For example, with respect to the activity of antitumor antibodies targeting liver-resident tumor cells, it was suggested that Kupffer cells and FcγRIV in combination with FcγRI play a crucial role in the activity of the cytotoxic antibody [57]. Indeed, our results demonstrate that FcγRIV with about 50,000 ABC is the most abundant FcγR on Kupffer cells. In combination with a low level of the inhibitory FcγRIIb expression, this suggests that Kupffer cells may be very efficient in taking up opsonized materials from the blood. Evidence along these lines has been provided by studies demonstrating that KC play an important role in the FcγR-mediated clearance of circulating antigen–antibody conjugates or larger immune complexes (see, e.g., Reference [58]) or by intravital imaging studies showing that KC participate in the depletion of opsonized peripheral blood B cells in the liver [7]. In addition, the autoantibody-mediated removal of red blood cells by liver macrophages has been demonstrated directly via the histological detection of iron deposits in Kupffer cells in vivo [59].

Another prime example for an organ where FcγRs on macrophages are of major importance is the lungs. We showed that the antibody-dependent killing of melanoma metastasis is strictly dependent on FcγRI and FcγRIV [40,60]. A more recent study further suggested that alveolar macrophages may be key to the killing of tumor cells via ADCC in the lungs, which is in line with the high expression of both receptors on alveolar, but not on interstitial, macrophages (Table 1) [6]. Apart from tumor-specific antibodies, it was demonstrated that broadly neutralizing antibodies (bNAbs) against Hemagglutinin (HA) require Fc–FcγR interactions for the optimal protection of mice from influenza virus infection in vivo [61]. Subsequently, this protective effect of bNAbs was attributed largely to AM, and it was demonstrated that both murine and human non-neutralizing antibodies (nonNAbs) against HA are dependent on AM and require antibody–FcγR interactions to optimally protect mice from severe disease [62]. Moreover, the protection of mice against the influenza A virus by immune serum against the ectodomain of matrix protein 2 (M2e) depends on alveolar macrophages and FcγRs [63]. Thus, the deficiency of both FcγRI and FcγRIII strongly reduced the protection by anti-M2e serum (containing a range of IgG subclasses) against the influenza virus challenge. Purified anti-M2e antibodies of the IgG1 subclass were specifically dependent on functional FcγRIII to confer an effective protection [63]. Along the same lines, mice in which alveolar macrophages were depleted by the nasal instillation of clodronate liposomes failed to achieve the antibody-mediated elimination of SARS-CoV [64].

In the skin, various lines of evidence indicate that the autoantibody-dependent skin blistering disease epidermolysis bullosa acquisita (EBA) is dependent on antibody–FcγR interactions (reviewed in Reference [65]). Specifically, FcγRIV has been identified as the key mediator of tissue injury in antibody transfer-induced experimental EBA. Regarding the involved effector cells, it has been suggested that, besides the well-known function of neutrophils in pathogenesis, monocytes/macrophages also directly contribute to blister formations in experimental EBA [66]. As FcγRIV, however, is virtually absent from dermal macrophages, it seems likely that blood-derived monocytes that can increase FcγRIV expression upon activation [6] may be the key effector cells here. Interestingly, FcγRIIb has recently been shown to control skin inflammation in an active model of EBA. While this effect has been mainly attributed to neutrophils as the main effector cells in EBA [67], the extremely high expression of the inhibitory receptor by DM may also suggest that DM could also be involved in modulating EBA. Indeed, apart from neutrophils, monocytes/macrophages have been suggested as additional effector cells in skin blister formations [66].

In the kidneys, CX3CR1^high^ F4/80^high^ kidney-resident macrophages are located at the interstitium between endothelial cells of peritubular capillaries and the basement membrane. Recently, Stamatiades et al. showed that renal macrophages rapidly take up circulating immune complexes that reach the interstitium via transendothelial transport [68]. The activation of these macrophages induces inflammation, which is probably the mechanism underlying the particular susceptibility of the kidneys to developing inflammation due to circulating immune complexes. Stamatiades et al. further identified FcγRIV as the key receptor for the renal macrophage response to circulating IC [68]. In a similar manner, Kaneko and colleagues demonstrated that FcγRIV was critical for IC-induced kidney inflammation in a model of nephrotoxic nephritis [69]. The important role of FcγRIV in antibody-dependent kidney pathology is in line with our finding that FcγRIV with more than 100,000 ABC is by far the most abundant Fcγ receptor on kidney-resident macrophages. In fact, FcγRIV expression on kidney-resident macrophages is the highest among all the organ-resident macrophage subsets investigated in this work. Finally, the results by Chauhan et al. on the expression of FcγRs on microglia in a steady state revealed a high expression of inhibitory FcγRIIb and barely any expression of FcγRIV [70]. This corresponds well with our quantitative data with 55,000 ABC for FcγRIIb but only 3000 ABC for FcγRIV.

### 3.2. Comparison to FcγR Expression on Human Tissue Macrophage Subsets

A recent study by Bruggeman and colleagues semiquantitatively assessed the FcγR expression on human ex vivo differentiated macrophages and select human tissue macrophages, including alveolar macrophages and macrophages from the spleen and liver [71]. Even though the methodology did not enable a quantification and direct comparison of the expression levels of different FcγRs, the relative expression of each receptor on cells from different tissues could be compared and revealed striking differences between the different macrophage populations. Importantly, this set of data corresponds well with several of our observations in the murine system. Thus, in humans and mice, FcγRI revealed a much higher expression on alveolar macrophages than on most macrophages in the spleen (i.e., red pulp macrophages) and liver. Additionally, for murine FcγRIV, we found a higher expression on AM than on Kupffer cells and red pulp macrophages, which both revealed comparable expressions. Accordingly, Bruggemann et al. found a very similar expression pattern for FcγRIIIA, the human ortholog of murine FcγRIV [27]. The same is true for the prominent expression of murine FcγRIII and the human ortholog FcγRIIA on AM. Additionally, human splenic CD169^+^ macrophages were found to exclusively express FcγRIIa [72], in line with our finding that the murine ortholog FcγRIII was the most prominently expressed Fcγ receptor on marginal macrophages in mice, whereas FcγRI and FcγRIIb are largely missing. Human red pulp macrophages predominantly expressed low-affinity receptors FcγRIIa and FcγRIIIa, did not express inhibitory FcγRIIb and had very low levels of high-affinity receptor FcγRI [72]. Here, we found a very similar expression pattern for the mouse ortholog Fc receptors III and IV on murine RPM and low FcγRIIb expression, albeit with a somewhat more prominent FcγRI expression. However, there were also some differences between mice and humans, especially with respect to the inhibitory FcγRIIb. Thus, among the macrophage populations tested by Bruggeman et al., including alveolar macrophages and macrophages from the spleen and liver, the highest expression of FcγRIIb was found on Kupffer cells. In contrast, in mice, we found the most prominent expression among these macrophage populations on alveolar macrophages and a significantly lower expression on KC.

In summary, our work emphasizes the high level of diversity of FcγR expression patterns on diverse subsets of tissue-resident macrophages during a steady state. Potential sex differences in the baseline FcγR expression on tissue-resident macrophages and, in particular, the question if and how this baseline expression changes during macrophage development or during tissue inflammation/repair will be interesting topics for future studies.

## 4. Material and Methods

### 4.1. Mice

Female mice at 8–14 weeks of age on C57BL/6 background were used in all the experiments. C57BL/6J mice (JAX strain 000664) were purchased from Janvier (Le Genest-Saint-Isle, France). For the identification of cells that actively expressed CX3CR1, we used B6 *cx3cr1^+/gfp^* mice in which the *cx3cr1* gene was replaced by the gene encoding green fluorescent protein (GFP) on one allele. In order to identify cells that at any point of their development also transiently expressed CX3CR1, we used BL6 Rosa26-*td tomato* x BL6 *cx3cr1*-*cre* mice. These mice express cre recombinase under control of the *cx3cr1* promoter and have a floxed STOP cassette in front of the *td tomato* gene. Mice were kept in the animal facilities of Friedrich–Alexander University Erlangen–Nürnberg, Bavaria, Germany under specific pathogen-free conditions in individually ventilated cages, according to the guidelines of the National Institutes of Health and the legal requirements in Germany. Animal experiments conducted in the animal facility of the FAU were approved by the government of lower Franconia.

### 4.2. Preparation of Murine Tissue Macrophages

To isolate tissue-resident macrophages from various organs, mice were euthanized and perfused with PBS to minimize the presence of residual blood cells. After bronchoalveolar lavage for the isolation of alveolar macrophages, the lung, liver, spleen and ears were harvested from C57Bl/6J mice. For the characterization of microglia in the brain and kidney-resident macrophages, we accordingly used B6 *cx3cr1^+/gfp^* mice to utilize the pronounced CX3CR1 expression of these cells for their identification.

The different organs/tissues were subjected to enzymatic digestion using DNase I, Collagenase D and/or Dispase^®^, followed by organ-specific isolation protocols. These protocols are presented in the supplementary methods for each macrophage population/organ.

Finally, the cells were resuspended in cold FACS buffer containing sodium azide to inhibit changes in the surface presentation of the proteins and were processed for the flow cytometric analysis.

### 4.3. Flow Cytometric Characterization of Murine Tissue Macrophages

Single-cell suspensions were incubated for at least 10 min on ice with Fc block antibodies to minimize unspecific binding to Fc receptors, followed by staining with fluorochrome-conjugated antibodies for at least 20 min. For Fc-blocking, the cells were pretreated with anti-CD16/32 clone 2.4G2 to specifically block FcγRIIb and III only when FcγRIV was quantified. This Fc block was not used in the analysis of FcγRI, since 2.4G2 may also block high-affinity receptor FcγRI via its Fc part on cells where 2.4G2 antibody was bound in *cis* to FcγRIIb/III [73]. Since we also found that medium-affinity receptor FcγRIV can cause false-positive results in flow cytometry by binding to the Fc part of several rat and mouse IgG subclasses [74], FcγRIV was blocked by clone 9E9 in all the assays where other receptors than FcγRIV were to be analyzed.

The antibodies used for the identification of the macrophage populations are listed in Appendix A.

A common initial gating step prior to the specific gatings depicted in Figure 3, Figure 4, Figure 5, Figure 6, Figure 7 and Figure 8 was the exclusion of the cell aggregates based on their light scatter characteristics by plotting the amplitude of the forward light scatter (FSC-H) against the corresponding area under the curve_signal vs. time_ (FSC-A). To exclude dead cells, we used 4′,6-diamidino-2-phenylindol (DAPI), which penetrates dead cells much more easily than viable cells and is highly fluorescent upon intercalation into DNA. The presence of autofluorescent particles like debris and distinct cells causes “tailing” and the diagonal appearance of the respective poulations of the events in dot plots. This has to be taken into account upon the gating of cell populations and often requires the usage of polygonal gates rather than the simple distinction between “positive” and “negative” events based on a single threshold value.

Since pronounced autofluorescence is not unusual among macrophages, the apparent fluorescence for any antigen stained by fluorescent antibodies may be caused by specific antibody binding or by autofluorescence or a combination of both. To rule out the possibility that the observed apparent expression of a critical marker is due to autofluorescence, we propose that fluorescence minus one (FMO) controls be performed for each marker that is critical for identifying the distinct macrophage populations. For FMO, the control cells are stained with the respective panel of antibodies for the identification of a particular cell population, except the one directed against the antigen of interest. Here, we did this in pretests for important markers like, e.g., Langerin, F4/80, MerTK and MHC-II.

### 4.4. Bright Field Microscopy

For the light microscopic characterization of their morphology, some cell populations were sorted according to their flow cytometric characteristics described above using a FACSAria III cell sorter (BD Biosciences, Heidelberg, Germany). Subsequently, the cells were mounted on microscope slides by CytoSpin centrifugation (Cellspin I, Tharmac GmbH, Waldsoms, Germany) and stained with a Hemacolor^®^ Azur/Eosin kit (Sigma-Aldrich/ Merck KGaA, Darmstadt, Germany) prior to analysis with a Axio Observers 7 inverted microscope (Carl Zeiss Microscopy GmbH, Oberkochen, Germany).

### 4.5. Fluorescence Microscopy

To characterize the tissue location and receptor expression of spleen macrophages, the spleen was excised from perfused mice, embedded in the TissueTec O.C.T. compound (Sakura Finetek Germany GmbH, Umkirch, Germany) and frozen on dry ice. Samples were cut into 5-µm sections using a Microm HM550 cryostat (Thermo Fisher Scientific Germany, Braunschweig, Germany), and the sections were fixed with acetone and dried. Prior to staining, the sections were blocked with 5% goat serum in PBS for 30 min at room temperature and then stained with antibodies against the indicated antigens for 60 min at RT. Fluorescence microscopy of the stained samples was performed with an Axio Observers 7 inverted microscope.

### 4.6. Quantification of Fcγ Receptors

Quantification of the Fcγ receptors was carried out as described in Reference [42]. Briefly, the antibody-binding capacities (ABC) for antibodies against the respective Fcγ receptors were calculated using reference curves that correlated with the median fluorescence intensity (MFI) of a cell population upon binding by the respective fluorochrome-conjugated anti-FcγR antibody, with the antibody-binding capacity (ABC) represented as the “number of antibody-binding sites/cell”. These curves were generated using commercially available Quantum Simply Cellular (QSC) microspheres (Bangs Laboratories Ltd., Fishers, IN, USA), with a known numbers of antibody-binding sites as provided by the manufacturer. Beads and cells were stained with the same concentrations of the respective anti-FcγR antibodies, using concentrations that saturated the specific antibodies binding to both cells and beads as completely as possible. The following PE-conjugated anti-FcγR antibodies were used for mouse receptor quantification: anti-msFcγRI/CD64 mouse IgG1 clone X54-5/7.1, anti-msFcγRIIb/CD32b mouse IgG2a clone AT130.2, anti-msFcγRIII/CD16 rat IgG2a clone 275003 and anti-msFcγRIV Arm. hamster IgG clone 9E9 (see, also, Appendix A).

We used a single type of fluorophore—Phycoerythrin (PE)—to minimize the potential systematic variations in the quantification of different receptors by the engagement of different fluorophores. PE was chosen, since, due to its high molecular weight, typically only a single fluorochrome molecule is conjugated to each antibody, thereby minimizing the variations in a specific fluorescence. In addition, PE lacks the pronounced self-quenching capacity of fluorochromes like FITC [75].

According to the host species of the respective anti-FcγR antibody, anti-mouse IgG or anti-rat IgG-specific QSC beads were used following the manufacturer’s instructions. Since anti-FcγRIV is derived from the Armenian hamster, for which no QSC beads are available, anti-mouse IgG beads were precoated with mouse anti-hamster antibody (see the description in Reference [42]). To be able to subtract the ABC background values corresponding to the background fluorescence of the respective cells, we used “fluorescence minus one” (FMO) controls in each experiment. Therefore, the cells were stained with a respective panel of antibodies for the identification of that particular cell population, except anti-FcγR antibodies. Of note, due to the use and quantification of the FMO controls, pronounced autofluorescence of macrophages is also not problematic for receptor quantification, as it is present in both FMO and anti-FcγR-stained samples and is thus included in the subtracted ABC background values.

The flow cytometric analysis was performed on FACS Canto II (BD Biosciences).

An example for the quantification procedure from the flow cytometric analysis of the reference beads to the generation of the reference curve for the calculation of the ABC from the MFI is provided in Figure 2C.

For the analysis of the macrophages from the skin, liver, lung and spleen, each single B6 mouse was used and analyzed in an independent experiment in one day. For the analysis of the microglia and kidney-resident macrophages, groups of two mice were analyzed per day independent from the others. The reference curves were newly prepared for two groups of mice, respectively.

### 4.7. Software

The flow cytometric data were analyzed with FACSDiva V6 (BD Biosciences) and Flowlogic (Miltenyi Biotec, Bergisch-Gladbach, Germany). QuickCal^®^ software, provided by Bangs Laboratories Ltd., was used for the ABC calculations. The figures were created using PRISM V9.2 (GraphPad Software, San Diego, CA, USA) and BioRender.com. ZEN software (Carl Zeiss AG, Oberkochen, Germany) was used for the bright field and fluorescence microscopy.

## Figures and Tables

**Figure 1 ijms-22-12172-f001:**
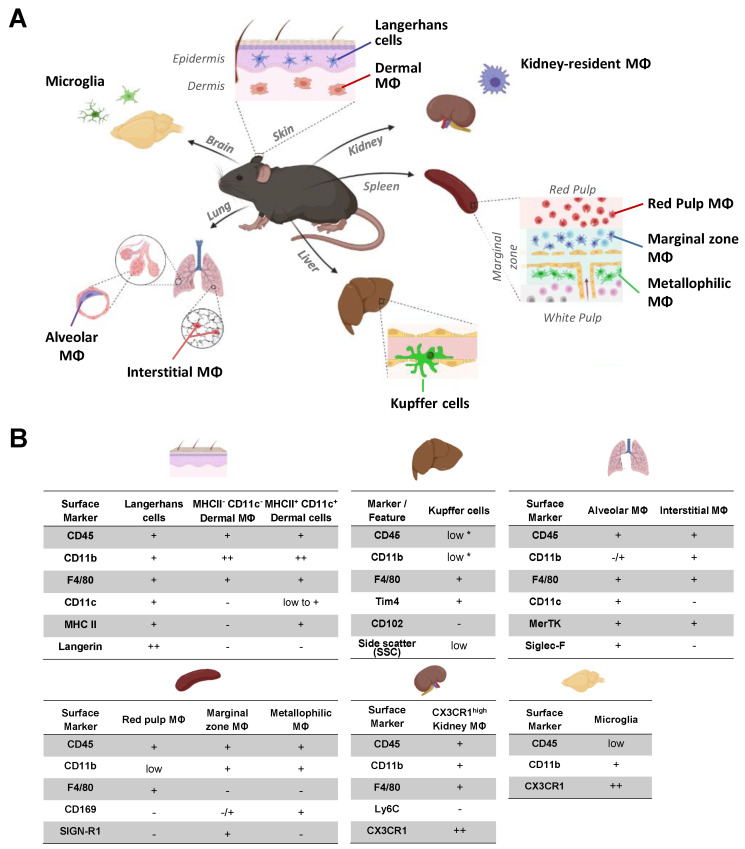
Markers allowing to identify tissue-resident macrophages subsets. (**A**) Schematic representation of the organs and organ-specific tissue-resident macrophages analyzed in this work, i.e., microglia from the brain; Langerhans cells from the epidermis and dermal macrophages from the dermis of the skin; kidney-resident macrophages, red pulp macrophages, the marginal zone and metallophilic macrophages from the spleen; Kupffer cells from the liver, alveolar macrophages from bronchoalveolar lavage and interstitial macrophages from tissue of the lung. (**B**) Depicted are the cell surface markers and cellular features of tissue-resident macrophages that were used for the identification of the respective macrophage population(s) by flow cytometry: no clear staining with the respective antibody; −/+, no clear shift upon staining with the respective antibody but “broadening” of the cell population in the corresponding dot plot; low, a distinct but minor shift upon staining with the respective antibody or low light scatter in comparison to several other cell populations; +, a distinct shift upon staining; ++, very prominent endogenous fluorescence (CX3CR1-GFP) or increase in fluorescence upon staining (Langerin, CD11b) in comparison to the other cell populations and *, low staining in comparison to fluorescence minus one control but adds to the pronounced autofluorescence, resulting in a high apparent fluorescence in total.

**Figure 2 ijms-22-12172-f002:**
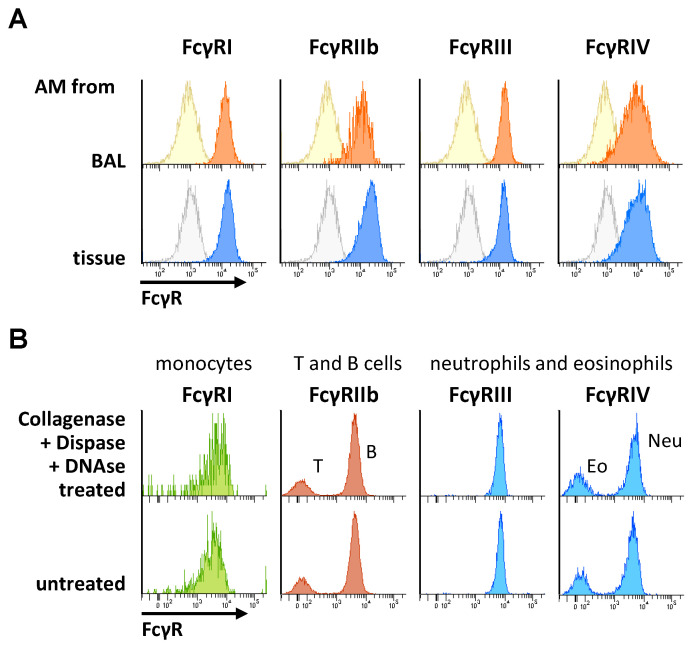
Impact of enzymatic digestions on FcγR expression analysis. (**A**) Depicted are the flow cytometric analyses of alveolar macrophages isolated by BAL (upper panels) or in enzymatically digested lung tissue (lower panels). Fluorescence intensities are shown for a respective FMO control (light ocher and light-blue histograms) or for cells stained with PE-conjugated antibodies directed against the indicated FcγRs. X-axis scaling for the BAL samples is identical to the depicted scaling of the tissue samples (**B**) Histograms showing the fluorescence intensities of the indicated murine peripheral blood leukocytes upon staining with PE-conjugated antibodies that are specific for the indicated FcγRs. The blood samples have been incubated without enzymes (lower panels) or with collagenase D, Dispase^®^ and DNase I (upper panels) prior to antibody staining. X-axis scaling for the enzymatically treated samples is identical to the depicted scaling of the untreated samples. (**C**) Depicted is—as an example—the establishment of an anti-CD64 (FcγRI) reference curve for the deduction of the number of anti-FcγR-binding sites presented as the antibody-binding capacity (ABC) from the median fluorescence intensity. Five different types of QSC beads with known anti-mouse IgG-binding sites were loaded with an anti-FcγR antibody and analyzed by flow cytometry. Aggregated beads were excluded, and the fluorescence intensity of each single bead population was measured (upper panel). In the lower left panel, an excerpt from a QuickCal™ calculation sheet is shown, in which the detection threshold and quality of fitting (represented by the regression coefficient) are calculated from the known ABC values of the beads and measured fluorescence intensity (“channel”). The lower right panel depicts the reference curve for CD64 fitted to the anti-CD64-binding capacity (ABC) vs. fluorescence intensity (histogram channels) of the reference beads.

**Figure 3 ijms-22-12172-f003:**
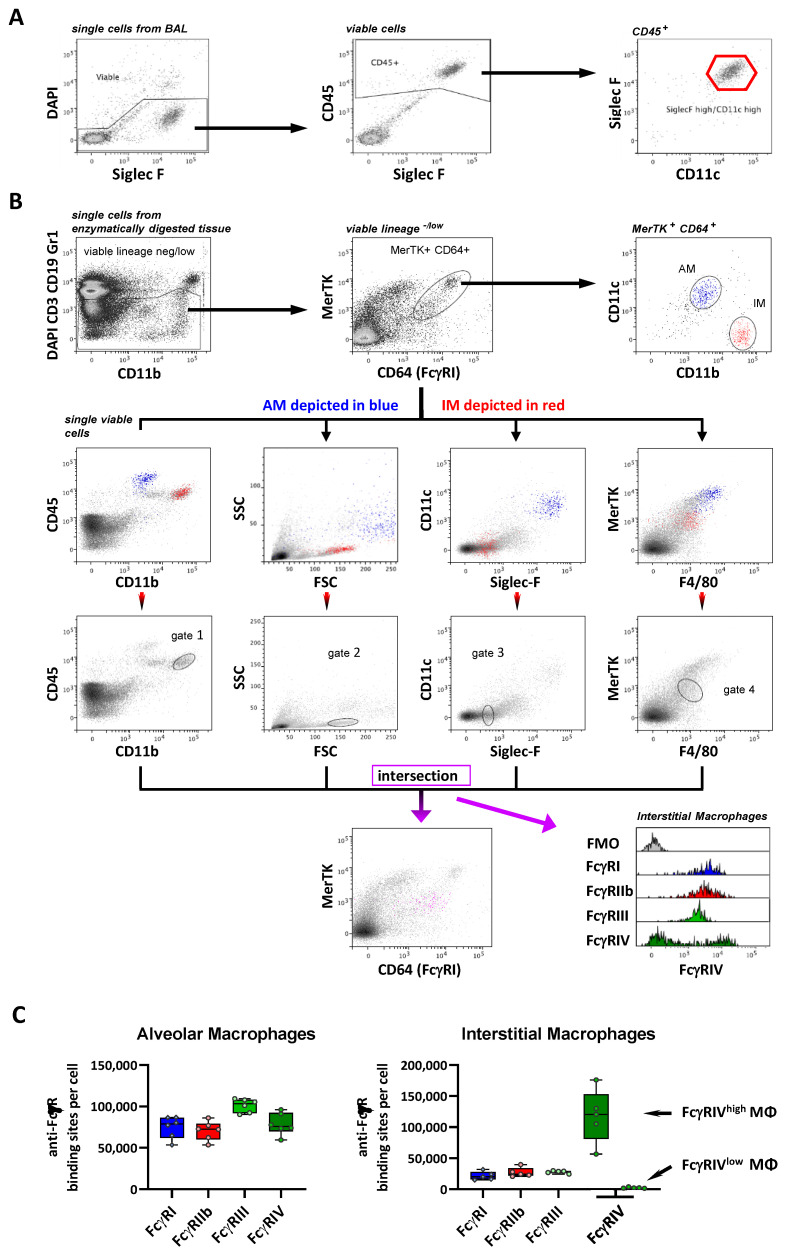
Characterization of the FcγR expression on pulmonary macrophages. (**A**) Depicted are single cells from the bronchoalveolar lavage (BAL). By selecting DAPI-negative cells, dead cell exclusion was performed. Next, the viable cells were subdivided based on their expression of CD45, Siglec F and CD11c. Among the CD45-positive populations, alveolar macrophages were identified by a pronounced expression of both Siglec F and CD11c. (**B**) Single cells from enzymatically digested lung tissue were stained with DAPI and antibodies specific for B-cell (CD19) and T-cell (CD3) markers, as well as Gr-1 (Ly6G and Ly6C) in a dump channel. Viable and CD19/CD3-negative cells and CD11b-positive Gr-1-negative-to-low cells were analyzed further to identify the macrophages as MerTK^+^ CD64^+^ cells. The IM were characterized as CD11b^high^ CD11c^−/low^, shown in red, and AM as CD11b^low^ CD11c^high^ in blue for comparison (upper panels). The IM and AM are marked in red and blue in dot plots for the various combinations of markers (second row of panels). The gates were set to identify IM in these dot plots (third row of panels). The intersection of these gates, depicted in magenta, was used to define a population that corresponds to the CD64^+^ MerTK^+^ IM in the anti-CD64-stained sample (lower panel). (**C**) Depicted are the number of anti-FcγR-binding sites per cell as a quantitative correlate of the FcγR receptor expression. Data are presented as box plots showing the median and interquartile range and whiskers showing extremes together with all single values. *n* = 6 for AM and 5 for IM.

**Figure 4 ijms-22-12172-f004:**
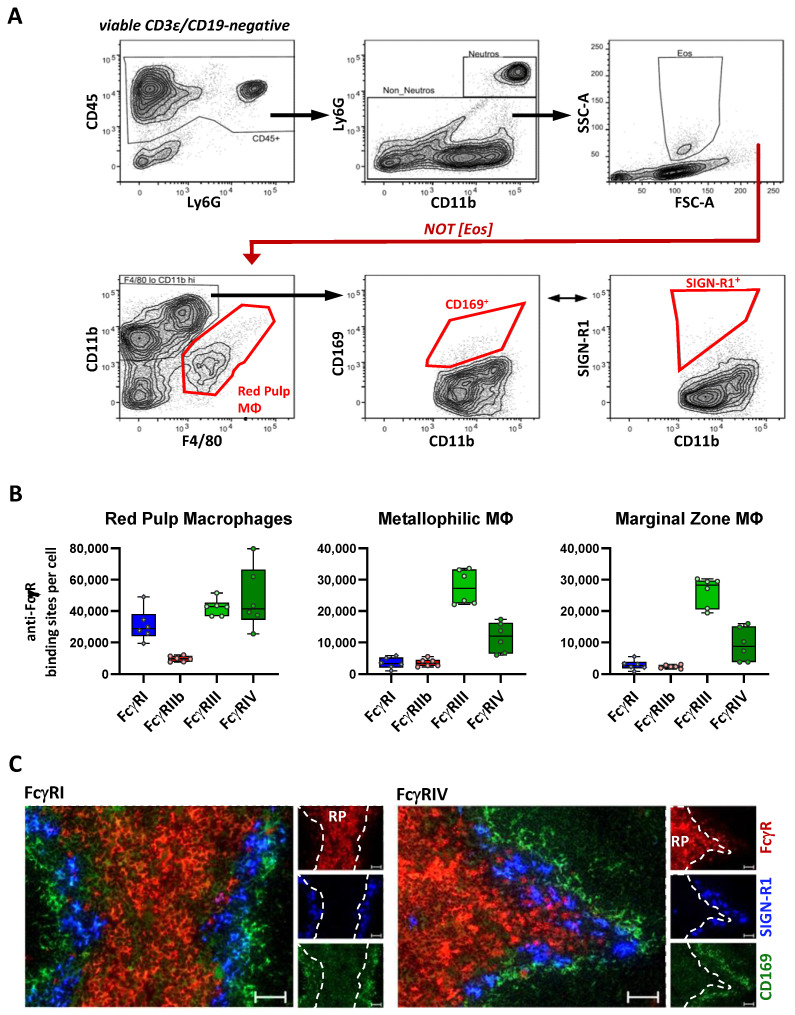
Characterization of the FcγR expression on splenic macrophages. (**A**) Depicted is the gating strategy for splenic macrophages. B, T and dead cells were excluded by using a dump channel (using CD19 and CD3e and DAPI). From CD45-positive leukocytes, neutrophils were excluded by Ly6G expression and eosinophils by their light scatter characteristics with high side scatter (SSC). Within the remaining leukocyte population, red pulp macrophages were characterized by low CD11b expression and pronounced F4/80 expression. Among CD11b-high cells with no or low F4/80 staining, metallophilic macrophages were characterized by the expression of CD169. Cells with a high expression of SIGN-R1 were regarded as marginal zone macrophages. (**B**) Shown are the number of anti-FcγR-binding sites per cell as a quantitative correlate of the FcγR expression on RPM, the marginal zone and metallophilic MΦ. Data are presented as box plots showing the median and interquartile range and whiskers presenting extremes with all single values. *n* = 6. (**C**) Analysis of the FcγRI and RIV expression on splenic macrophages by fluorescence microscopy. FcγRI (left panel) and FcγRIV (right panel) are shown in red. The presence of SIGN-R1 and CD169 is indicated by blue or green, respectively. Shown is the overlay of FcγR, SIGN-RI and CD169, as well as each single fluorescence channel. In the single-channel pictures, the position of the red pulp is shown based on its demarcation from the SIGN-R1-positive macrophages (dashed line). Scale bars represent 50 µm.

**Figure 5 ijms-22-12172-f005:**
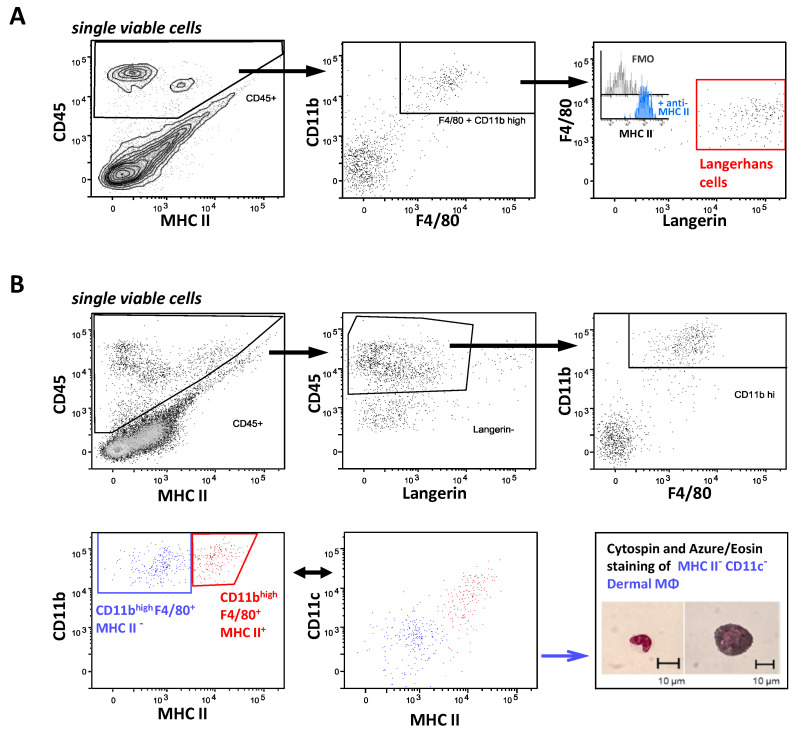
Characterization of the FcγR expression on skin macrophage subsets. (**A**) Shown is the gating strategy for the identification of Langerhans cells. CD45-positive populations among single viable cells from an enzymatically digested epidermis were examined for expression of the surface markers CD11b and F4/80. Among CD11b and F4/80-double-positive cells, Langerhans cells were identified by a pronounced expression of Langerin. The histogram inset in the right panel depicts their MHC II expression by comparing the fluorescence in the samples without (FMO, grey histogram) and with anti-MHC II antibody (blue histogram). (**B**) Gating strategy for Langerin-negative F4/80-positive dermal cells. Single viable cells from enzymatically digested and dermis-enriched skin were analyzed regarding their CD45 and Langerin expression to exclude Langerhans cells among the CD45-positive leukocytes. Among the CD45^+^ Langerin-negative cells, two subpopulations within the CD11b and F4/80-positive cells were distinguished based on a high or low level of MHC II expression (depicted in blue or red), respectively. Both population were then examined for CD11c expression. Inset: Microscopic analysis of the sorted MHC II^−^ CD11c^−^ DM. (**C**) Depicted is the ABC for each FcγR per cell as a quantitative correlate of the individual FcγR numbers. Data are presented as box plots showing the median and interquartile range and whiskers depicting extremes together with all single values. *n* = 6 or 5 upon the exclusion of single outliers according to the Grubbs’ test with alpha = 0.1.

**Figure 6 ijms-22-12172-f006:**
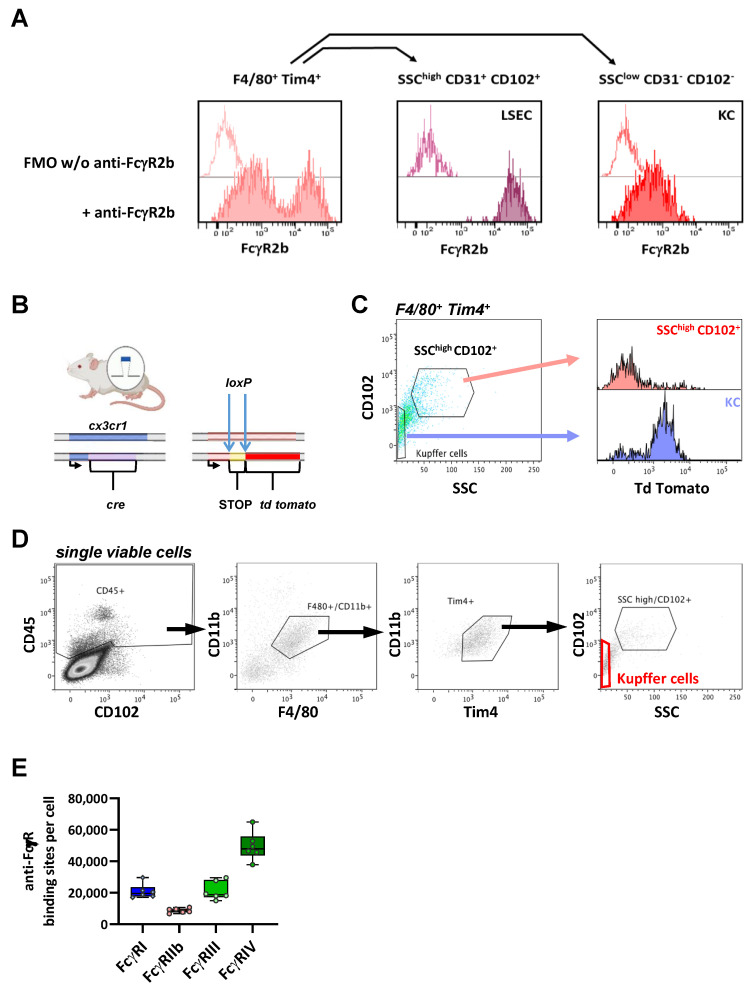
Characterization of FcγR expression on hepatic Kupffer cells. (**A**) Depicted is the fluorescence of F4/80 Tim4-double-positive liver cells with or without anti-FcγRIIb, respectively. Among these F4/80^+^ Tim4^+^ cells, FcγRIIb expression was analyzed in populations with either a high side scatter (SSC) and the presence of endothelial markers CD31 and CD102 (medium panel) or a low side scatter and the absence of CD31 and CD103 (right panel). (**B**) Genetic characteristics of BL6 Rosa26-*td tomato* x BL6 *cx3cr1-cre* mice. In these mice, one allele of the CX3CR1 locus mice encodes a Cre recombinase under control of the CX3CR1 promoter. In addition, one allele of the Rosa 26 locus contains a *td tomato* gene with a preceding floxed STOP cassette. (**C**) Depicted is the tdTomato expression in the F4/80^+^ Tim4^+^ subpopulations that were either SSC^high^ CD102^+^ or CD102-negative with a low light side scatter, which represents hepatic Kupffer cells. (**D**) Gating strategy for the identification of Kupffer cells. Among single viable cells from enzymatically digested liver, the CD45^+^ leukocytes were analyzed with respect to CD11b, F4/80 and Tim4 expression. Among the cells positive for all three markers, we characterized KC by their low side scatter and absence of endothelial marker CD102. (**E**) Depicted is the ABC for each FcγR per cell as a quantitative correlate of the individual FcγR numbers. Data are presented as box plots depicting the median and interquartile range and whiskers showing extremes together with all the single values; *n* = 6.

**Figure 7 ijms-22-12172-f007:**
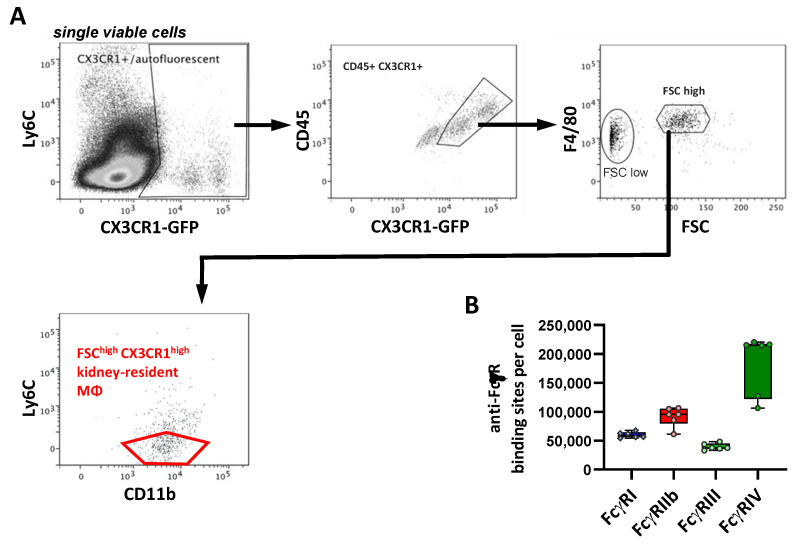
Characterization of FcγR expression on kidney-resident macrophages. (**A**) Gating strategy for the identification of kidney-resident MΦ. Single viable cells from enzymatically digested kidneys of B6 *cx3cr1^+/gfp^* mice were examined for *cx3rcr1* promoter-driven GFP and CD45 expression. CD45^+^ CX3CR1^+^ cells were then analyzed with respect to F4/80 expression and cell size, as reflected by their forward light scatter characteristics (FSC). In the F4/80^+^ FSC^high^ population, we then selected cells that were negative for Ly6C but positive for CD11b. (**B**) Depicted are the number of anti-FcγR-binding sites per cell as a quantitative correlate of FcγR expression. Data are presented as box plots depicting the median and interquartile range and whiskers showing extremes together with all single values; *n* = 6.

**Figure 8 ijms-22-12172-f008:**
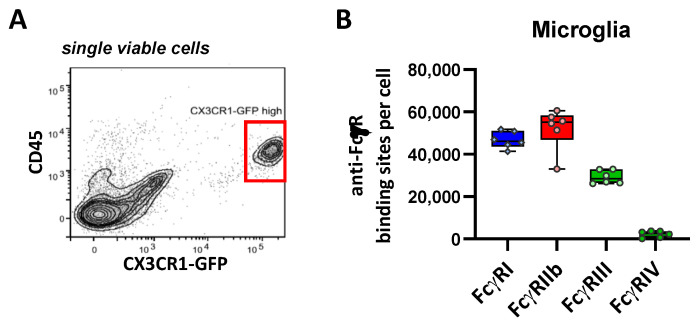
Characterization of FcγR expression on the microglia. (**A**) Flow cytometric characterization of microglia. Viable cells from a single cell suspension of the enzymatically digested and myelin-ablated brain of B6 *cx3cr1^+/gfp^* mice were examined for *cx3rcr1* promoter-driven GFP expression and CD45 expression. (**B**) Depicted are the number of anti-FcγR-binding sites per cell as the quantitative correlate of Fc receptor expression. Data are presented as box plots depicting the median and interquartile range and whiskers showing extremes together with all single values; *n* = 6.

**Table 1 ijms-22-12172-t001:** Expression of the Fcγ receptors on tissue-resident macrophages.

Median ABC (×10³ Binding Sites/Cell)	FcγRI	FcγRIIb	FcγRIII	FcγRIV
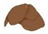	**Kupffer Cells**	20	9	19	48
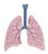	**Alveolar Macrophages**	79	73	103	76
**Interstitial Macrophages**	20	24	29	121/2 *
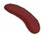	**Red Pulp Macrophages**	29	10	43	41
**Metallophilic Macrophages**	3	3	27	12
**Marginal Zone Macrophages**	3	2	28	9
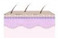	**Langerhans Cells**	<1	54	49	<1
**MHC II^−^ CD11c^−^ Dermal Macrophages**	99	624	135	<1
**MHC II^+^ CD11c^low to +^ Dermal Cells**	32	236	150	<1
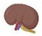	**Kidney-resident Macrophages**	59	95	38	215
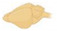	**Microglia**	46	55	28	3

(*) FcγRIV^high^/FcγRIV^low^ IM.

## Data Availability

The individual ABC values presented in this study are openly available in FigShare at doi 10.6084/m9.figshare.16961161.

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
