# Peer review of "There Is Strength in Numbers: Quantitation of Fc Gamma Receptors on Murine Tissue-Resident Macrophages"

_ijms, 2021, doi:10.3390/ijms222212172_

Round 1
Reviewer 1 Report
In this study author shows an different expression of FcγRs on the surface of organ resident macrophages. Although the study is descriptive, it is very interesting and complete. With a very detailed methodology. However some points can be improved to increase the quality of the study
In section 2.2 the author suggest that enzyme mix used for tissue resident macrophage isolation does not affect cellular FcγR expression. But in figure 2 the axis lines X and Y are not visible, therefore, it is impossible to verify the fluorescence intensitie. Please improve that . Moreover de the resolution of the figure2 is poor , impossible to see the legends of figure C or sheet of calculation , and some words are underlined in red
In section 2.3 FcγR quantification on lung macrophage subpopulations. The structure of the article is strange and the reviewer finds interesting results in the figure captions but not in the text of the result. Please highlight the important results kept in the text and not in the legend, Ex. in line 191 and 192 da page 7 “Based on fluorescence intensity in FMO samples and samples stained for the different FcγR, this population shows homogenous expression for FcγRI, IIb and III, but reveals two subpopulations with high or low FcγRIV expression (lower right panel)”. Please organize the article in such a way that the comments about the results are in the result part and that the legend of the figures is used only to understand the figure.
Author Response
Reviewers comment:
In this study author shows an different expression of FcγRs on the surface of organ resident macrophages. Although the study is descriptive, it is very interesting and complete. With a very detailed methodology.
Authors response:
We thank the reviewer for the appreciation of this work.
However some points can be improved to increase the quality of the study.
Authors response:
The reviewer pointed out some important points, whose revision significantly improved the quality of the manuscript.
In section 2.2 the author suggest that enzyme mix used for tissue resident macrophage isolation does not affect cellular FcγR expression. But in figure 2 the axis lines X and Y are not visible, therefore, it is impossible to verify the fluorescence intensities. Please improve that . Moreover de the resolution of the figure2 is poor , impossible to see the legends of figure C or sheet of calculation , and some words are underlined in red.
Authors response:
We now depict X-axis ticks for both the respective upper and lower panels and the numbering in the respective lower panels. In addition, we describe within the figure legend that in A) X-axis scaling for BAL samples is identical to the depicted scaling of tissue samples and that in B) X-axis scaling for enzymatically treated samples is identical to the depicted scaling of untreated samples.
In panel C we increased the size of several characters and numbers to enhance readability. In addition, we inserted Figure 2 (and all other figures) in a different format which enhances resolution and removes red spell-check lines.
In section 2.3 FcγR quantification on lung macrophage subpopulations. The structure of the article is strange and the reviewer finds interesting results in the figure captions but not in the text of the result. Please highlight the important results kept in the text and not in the legend, Ex. in line 191 and 192 da page 7 “Based on fluorescence intensity in FMO samples and samples stained for the different FcγR, this population shows homogenous expression for FcγRI, IIb and III, but reveals two subpopulations with high or low FcγRIV expression (lower right panel)”. Please organize the article in such a way that the comments about the results are in the result part and that the legend of the figures is used only to understand the figure.
Authors response:
We thank the reviewer very much for pointing out the incorrect figure legend. Due to unintended loss of a paragraph break a part of the main text had been assigned to the figure legend. This has now been corrected. In addition, we removed the phrase in original lines 191 and 192 mentioned by the reviewer and integrated the information into the corresponding main text (lines 217f in the revised version).
Reviewer 2 Report
Vorsatz et al. presented a descriptive study about the quantitation of FcγRs on mouse macrophages. Authors have used different tissue compartment, isolated tissue macrophages and measured FcγRs expression by either flow cytometry or immunohistochemistry. The study certainly has a merit and may aid future studies design to probe relative contribution of Fc gamma receptors on macrophage. However, the merits are severely masked, in this reviewer’s opinion, due to the presentation of generic data and shed limited light on the next aspect of FcγRs on murine tissue macrophage in the present form. Most, if not all, data presented in this manuscript could be harvested from the literature. This manuscript in a way gives in impressive of a review article with original data reproduced from the literature.
Major comments:
- Figure 1, if at all required, should be the last figure. It gives an impression of a review article.
- Several analyses presented in the study have been performed on different macrophage populations in previous studies. This study seems to be a compendium of previous studies in terms of macrophage Fc gamma receptor expression (doi.org/10.1016/j.molimm.2013.05.219; doi.org/10.1172/jci.insight.125503; org/10.4049/jimmunol.0903833; doi.org/10.1002/eji.200939884; doi.org/10.1172/jci25536; doi.org/10.1172/jci16577; doi.org/10.1111/cei.12935; doi.org/10.1371/journal.pone.0110966; doi.org/10.1172/JCI36452; doi.org/10.1002%2Fart.38311). In fact, authors have presented similar analysis of Fig. 2B,C in their previous publication (doi.org/10.3389/fimmu.2020.00118).
- Summary of the work ‘our work emphasizes the high level of diversity of FcγR expression patterns on diverse subsets of tissue resident macrophages during the steady-state’ is very generic and doesn’t seem to add up to the literature. In fact, it is obvious from the literature!
- ‘A prerequisite to quantifying FcγR numbers, or more accurately the FcγR-specific antibody binding capacity (ABC; represented as antibody binding sites/cell), on tissue resident macrophage subsets is a reliable macrophage identification in tissue extracts of skin, lung, liver, spleen, kidney and brain via FACS analysis.’ What does this sentence mean? This reviewer couldn’t comprehend! This is the general theme of the writing, mostly long sentences and loose the context through the process!
Minor comments:
- What does this title even mean in context of the study (episode 2)? There was not even a single mention of the title words or an explaining in the text!
- Please remove results from the figure legends “Using this gating strategy the quantification of FcγR numbers on pulmonary MΦ subsets demonstrated a pronounced heterogeneity with respect to FcγR numbers. Whereas, AM showed a strong expression of all four FcγRs at 75 000- 10 000 ABC, with FcγRIII being expressed at the highest level, IM expressed FcγRsI, IIb, and III at much lower levels (roughly at 25 000 ABC). Of note, with respect to FcγRIV expression two IM subpopulations with either a very high (120 000 ABC) or very low/absent FcγRIV expression could be identified (Table 1 and Figure 3C), consistent with the previous identification of several subsets of IM being present in the lung [44]” These sentences don’t belong to the figure legends.
Author Response
Reviewers comment:
Vorsatz et al. presented a descriptive study about the quantitation of FcγRs on mouse macrophages. Authors have used different tissue compartment, isolated tissue macrophages and measured FcγRs expression by either flow cytometry or immunohistochemistry. The study certainly has a merit and may aid future studies design to probe relative contribution of Fc gamma receptors on macrophage.
Authors response:
We thank the reviewer for acknowledging a merit of the data presented in this manuscript.
However, the merits are severely masked, in this reviewer’s opinion, due to the presentation of generic data and shed limited light on the next aspect of FcγRs on murine tissue macrophage in the present form. Most, if not all, data presented in this manuscript could be harvested from the literature. This manuscript in a way gives in impressive of a review article with original data reproduced from the literature.
Authors response:
Of course many groups work on Fc receptors on various cell types including macrophages. Accordingly, in the literature there are also many data on the expression of such receptors by macrophages, but the vast majority of results is either qualitative or semi-quantitative in nature. There are barely any data regarding actual Fc receptor numbers. We, thus aimed at providing such quantitative data with actual numbers which also allow for comparison of different receptors with each other and can be the basis for mathematical modelling. Therefore, we think that this work provides an additional value to the present literature.
Major comments:
1. Figure 1, if at all required, should be the last figure. It gives an impression of a review article.
Authors response:
We would like the keep the figure at this position in order to give the reader – especially also those which are no specialists in the topic of tissue macrophages – an introductory overview on the variety of tissue resident macrophages we analyzed in this work. In addition, we think that the listing of markers which we successfully used for identification of macrophage populations is a relevant information for interested readers.
2. Several analyses presented in the study have been performed on different macrophage populations in previous studies. This study seems to be a compendium of previous studies in terms of macrophage Fc gamma receptor expression (doi.org/10.1016/j.molimm.2013.05.219; doi.org/10.1172/jci.insight.125503; org/10.4049/jimmunol.0903833; doi.org/10.1002/eji.200939884; doi.org/10.1172/jci25536; doi.org/10.1172/jci16577; doi.org/10.1111/cei.12935; doi.org/10.1371/journal.pone.0110966; doi.org/10.1172/JCI36452; doi.org/10.1002%2Fart.38311).
Authors response:
The present work is clearly not a compendium of previous studies but the first one to generate quantitative expression data generated by the same methodology on a broad range of tissue macrophages in parallel. In the manuscripts mentioned by the reviewer expression data for Fc receptors on macrophages are typically presented by means of mRNA expression and especially of fluorescence intensity which is in fact a semi-quantitative approach. This is very valid for comparison of relative expression levels of the same receptor on different cells or under different physiological conditions, but due to the intrinsic differences in specific fluorescence of conjugated antibodies against different receptors (and even between different batches of antibodies against the same receptor) comparison between different receptors is hampered. In addition, the description of expression via fluorescence intensity without valid reference curves gives no hint on actual receptor numbers on the cell surface. Thus, in order to provide such quantitative data we performed the measurements presented here, which provide actual numbers and thus allow for comparison of different receptors with each other.
We included a paragraph acknowledging previous publications with qualitative and semi-quantitative data on FcγR expression by murine macrophages and include citations for those references provided by the reviewer, which refer to murine macrophages. In addition, we describe in somewhat more detail what we believe to be relevant differences between these approaches and the quantitative approach used here.
In fact, authors have presented similar analysis of Fig. 2B,C in their previous publication (doi.org/10.3389/fimmu.2020.00118).
Authors response:
In order to provide information of the methodology of receptor quantitation used here (and in the previous manuscript on FcγR quantities on peripheral blood leukocytes mentioned by the reviewer) we present an example for the generation of a reference curve in Figure 2C, as we did in our previous publication for the same reason. In the present Figure 2C of course we show primary data from the present experiments on tissue macrophages.
The results depicted in Figure 2B show samples of enzyme treated and non-treated cells in order to demonstrate that enzyme treatment, which is necessary for macrophage isolation from tissue samples, does not affect the staining for Fc receptors. This was no issue in our previous publication where we analyzed leukozytes in peripheral blood.
3. Summary of the work ‘our work emphasizes the high level of diversity of FcγR expression patterns on diverse subsets of tissue resident macrophages during the steady-state’ is very generic and doesn’t seem to add up to the literature. In fact, it is obvious from the literature!
Authors response:
Of course we never intended to give the impression that we were the first to show high diversity of FcγR expression patterns. That is why we choose the term “emphasizes”.
However, in contrast to most publications on receptor expression levels we present actual quantitative data for various macrophage populations, which we believe to provide an added value to the (mostly) semi-quantitative data showing unitless fluorescence intensities.
4. ‘A prerequisite to quantifying FcγR numbers, or more accurately the FcγR-specific antibody binding capacity (ABC; represented as antibody binding sites/cell), on tissue resident macrophage subsets is a reliable macrophage identification in tissue extracts of skin, lung, liver, spleen, kidney and brain via FACS analysis.’ What does this sentence mean? This reviewer couldn’t comprehend! This is the general theme of the writing, mostly long sentences and loose the context through the process!
Authors response:
The aim of this sentence was to introduce the reader to the importance of reliable methodology for isolation and characterization of macrophage populations. We acknowledge that our attempt to clarify the nature of the numbers presented here might be somewhat confusing when provided within that sentence. Thus, we shifted the phrase regarding ABC values to that paragraph where the first set of expression data are shown .
Minor comments:
- What does this title even mean in context of the study (episode 2)? There was not even a single mention of the title words or an explaining in the text!
Authors response:
We agree with the reviewer that this term may be confusing for the reader. During the process of writing we have reduced initial references to our previous publication which we had regarded as the first episode of our quantitative FcR description. Thus, we now removed the term “Episode 2” from the title.
2. Please remove results from the figure legends “Using this gating strategy the quantification of FcγR numbers on pulmonary MΦ subsets demonstrated a pronounced heterogeneity with respect to FcγR numbers. Whereas, AM showed a strong expression of all four FcγRs at 75 000- 10 000 ABC, with FcγRIII being expressed at the highest level, IM expressed FcγRsI, IIb, and III at much lower levels (roughly at 25 000 ABC). Of note, with respect to FcγRIV expression two IM subpopulations with either a very high (120 000 ABC) or very low/absent FcγRIV expression could be identified (Table 1 and Figure 3C), consistent with the previous identification of several subsets of IM being present in the lung [44]” These sentences don’t belong to the figure legends.
Authors response:
We thank the reviewer very much for pointing this out. Due to unintended loss of a paragraph break, part of the main text had been assigned to the figure legend. This has now been corrected.
Reviewer 3 Report
The manuscript describes the expression levels of the activating and inhibitory Fc gamma receptors (FcgRs) on tissue-resident macrophages during homeostasis. Using beads with known antibody binding capacity, the authors provide a quantitative value for the expression of each FcgR that can be directly compared to other macrophage populations. The authors provide detailed methods for gating strategies and tissue digestion for each macrophage subset and summarize the data at the end in a table. The results show the variability of FcgR expression between the different macrophage populations which are likely related to the microenvironment and role of the cells. These results are important for understanding the interaction of these cells with antibodies to contribute to Fc effector functions.
This work is timely and thoughtfully executed. While outside the scope of this manuscript, evaluation of changes to FcgR expression on tissue-resident macrophages during various disease states will be critical. This work provides the necessary groundwork for future studies on inflammation and repair.
Major Comments: None
Minor comments:
- Multiple flow plots have tails on the populations suggesting there is an issue with compensation. Alternatively, this could be related to autofluorescence. Please check compensation is accurate. If accurate, please mention in the method section about gating out the shift from autofluorescence.
- Figures 1 and 2 have red lines under some words. This appears to be a spell-check line, but the words are not misspelled. Please fix.
- Figure 2C is blurry. Please include a higher resolution image. In Figure 2 legend (line 139), the panel listed for lung tissue and BAL are flipped.
- On line 159, the abbreviation “AMs” is used. Please define.
- In Figure 5, the gating scheme for Langerhans cells is shown, however, MHCII is not included. Was this marker included as indicated by Figure 1B?
- Only female mice were used in this study, please comment on potential sex differences with FcgR expression on tissue-resident macrophages.
Author Response
Reviewers comment:
The manuscript describes the expression levels of the activating and inhibitory Fc gamma receptors (FcgRs) on tissue-resident macrophages during homeostasis. Using beads with known antibody binding capacity, the authors provide a quantitative value for the expression of each FcgR that can be directly compared to other macrophage populations. The authors provide detailed methods for gating strategies and tissue digestion for each macrophage subset and summarize the data at the end in a table. The results show the variability of FcgR expression between the different macrophage populations which are likely related to the microenvironment and role of the cells. These results are important for understanding the interaction of these cells with antibodies to contribute to Fc effector functions.
This work is timely and thoughtfully executed. While outside the scope of this manuscript, evaluation of changes to FcgR expression on tissue-resident macrophages during various disease states will be critical. This work provides the necessary groundwork for future studies on inflammation and repair.
Authors response:
We thank the reviewer very much for the appreciation of this work.
Minor comments:
1. Multiple flow plots have tails on the populations suggesting there is an issue with compensation. Alternatively, this could be related to autofluorescence. Please check compensation is accurate. If accurate, please mention in the method section about gating out the shift from autofluorescence.
Authors response:
Yes, the observed effects in the plots are based on autofluorescence (which is identified in unstained samples). We added a paragraph regarding the issue of autofluorescence in the section on flow cytometry (lines 570ff) and also added a sentence in section 4.6, describing why autofluorescence of macrophages does not interfere with Fc receptor quantification.
2. Figures 1 and 2 have red lines under some words. This appears to be a spell-check line, but the words are not misspelled. Please fix.
Authors response:
We thank the reviewer very much to point to these spell-check lines. We now inserted Figure 2 (and all other figures) in a different format which enhances resolution and removes red spell-check lines.
3. Figure 2C is blurry. Please include a higher resolution image. In Figure 2 legend (line 139), the panel listed for lung tissue and BAL are flipped.
Authors response:
We now inserted Figure 2 (and all other figures) in a different format which enhances resolution. Thanks a lot for the hint on the flipped listing! In the figure legend for Figure 2 the listing of the panels for BAL and tissue digestion has been corrected.
4. On line 159, the abbreviation “AMs” is used. Please define.
Authors response:
AM as abbreviation for Alveolar Macrophages has now been defined upon first mention of the term (line 134)
5. In Figure 5, the gating scheme for Langerhans cells is shown, however, MHCII is not included. Was this marker included as indicated by Figure 1B?
Authors response:
MHC II marker staining is depicted as the X-axis in Figure 5A, left panel, but – as the reviewer correctly pointed out - has not been shown as an identification marker in this panel. We now added as an inset a histogram in Figure 5A (right panel) including an FMO and antibody stained sample for MHC II.
6. Only female mice were used in this study, please comment on potential sex differences with FcgR expression on tissue-resident macrophages.
Authors response:
The reviewer raises an important point. We now added a hint to this potential issue. (lines 516f)
Round 2
Reviewer 2 Report
The authors have satisfactorily addressed most of the concerns of this reviewer. However, results of the experiments are still presented in the figure legends, which should be avoided. Eg. Fig. 7 legend 'CD45 and GFP double-positive cells revealed two populations which were both F4/80 positive but differed prominently in cell size as reflected by their forward light scatter characteristics (FSC). And 'The population with larger cell size (FSChigh) revealed higher fluorescence with respect to GFP as well as staining for CD11b, CD45 and F4/80 and for all Fcγ receptors upon subsequent FcR expression analysis (not shown). Because of their very small size and the fact that they mirrored the antibody-staining pattern of the FSChigh population but with lower fluorescence intensities, we considered the CD45+ CX3CR1-GFP+ CD11b+ F4/80+ Ly6C- FSChigh cells to be kidney resident macrophages and assumed that the FSClow population may represent subcellular particles thereof, which were not analyzed further. Similarly, in Fig. 8 'Microglia revealed intermediate CD45 expression and very high cx3cr1-associated GFP fluorescence. The also displayed pronounced expression of CD11b (not shown).
Please check the legends of all figures and present results in Results section for consistancy, which will help the reader.
Author Response
Followowing the hint of the reviewer on Figures 7 and 8 we identified presence of some results also in legends to Figures 4, 5, and 6. As suggested by the reviewer we removed these from figure legends and inserted the respective information in the corresponding main text as far as these results were not already described there in detail. In the legend to Figure 3 two marginal text modification were performed to clarity that these passages were not intended to describe results.